# Numerical Simulation Three-Dimensional Nonlinear Seepage in a Pumped-Storage Power Station: Case Study

**Shaohua Hu [1,2], Xinlong Zhou [1,*], Yi Luo [2] and Guang Zhang [1]**

[1]   School of Resource and Environment Engineering, Wuhan University of Technology, Wuhan 430070, China; sh_kxin@whu.edu.cn (S.H.); gzhang58@163.com (G.Z.)

[2]   Hubei Key Laboratory of Roadway Bridge and Structure Engineering, Wuhan University of Technology, Wuhan 430070, China; yluo@whut.edu.cn

[*]   Correspondence: xlzhousse2017@whut.edu.cn; Tel.: +86-27-87212127

**Abstract:** Due to high water pressure in the concrete reinforced hydraulic tunnels, surrounding rocks are confronted with nonlinear seepage problem in the pumped storage power station. In this study, to conduct nonlinear seepage numerical simulation, a nonlinear seepage numerical model combining the Forchheimer nonlinear flow theory, the discrete variational inequality formulation of Signorini's type and an adaptive penalized Heaviside function is established. This numerical seepage model is employed to the seepage analysis of the hydraulic tunnel surrounding rocks in the Yangjiang pumped-storage power station, which is the highest water pressure tunnel under construction in China. Moreover, the permeability of the surrounding rocks under high water pressure is determined by high pressure packer test and its approximate analytical model. It is shown that the flow in the surrounding rocks is particularly prone to become nonlinear as a result of the high flow velocities and hydraulic gradients in the nearby of the seepage-control measures and the high permeability fault. The nonlinear flow theory generates smaller flow rate than the Darcy flow theory. With the increase of nonlinear flow, this observation would become more remarkable.

**Keywords:** pumped-storage power station; nonlinear seepage; finite element method; variational inequality; seepage-control measures

## 1. Introduction

Electricity demand in China has risen speedily and reached an unprecedented level due to the rapid economic growth and modern development [1]. In China, primary electricity sources include the fossil fuel, hydropower, nuclear, and other renewable energy sources (e.g., wind, solar, biomass, wave and geothermal energy), which have generated total power capacity 5649.6 TWh in 2015 [2,3]. Considering the economic, technical and environmental benefits of the hydropower, most countries give priority to its development. China has the richest water resource in the world, with maximum exploitable hydropower resources of 694 GW and currently economic exploitable potential 380–400 GW [4]. However, due to the unbalanced hydropower distribution and economic development in China, the provinces in Southeast China, including Guangdong, Shanghai, Jiangsu, Zhejiang, Anhui, Fujian, Jiangxi and Shandong, accounted for more than 50% of electricity consumption in China, while Southwest China has the most fruitful hydropower resources in this country, which includes four provinces: Sichuan, Yunnan, Tibet and Guizhou. Pumped-storage power station offers a technically and economically feasible solution to the problem of the hydropower situation unbalance, which was used as early as 1890 in Italy and Switzerland, and promoted in the developed country [5]. For instance,

in the USA, Japan and the EU, the installed capacity of pumped-storage power reached 2.14%, 8.70% and 3.35% of their total installed capacity by 2010, respectively [6–8].

　　As shown in Figure 1, a typical pumped-storage power station usually contain a lower and an upper reservoir, a powerhouse and hydraulic tunnels. During off-peak electricity demand hours, the pumped-storage power station stores electricity by moving water from a lower to an upper reservoir [9–11]. Electrical energy is converted to potential energy and stored in the upper elevation. And during periods of peak hours, the stored water is released back through the turbines and converted back to electricity like a conventional hydropower station. In the abovementioned process, the amount of energy stored depends on the height difference between the two reservoirs and the total volume of water stored. As part of its energy transition strategy, China has accelerated the construction of pumped-storage power station since 2010 [12]. In the latest five year plan, China's government has set targets for pumped hydropower station capacity of 30 GW by 2015 to 70 GW by 2020, accounting for 3–5% of the total installed generation capacities in the country. As listed in Table 1, more than 10 pumped-storage power stations had been completed in the recent five years, such as: Bailianhe (1200 MW, Hubei), Baoquan (1200MW, Henan), Xianyou (1200 MW, Fujian), Qingyuan (1200 MW, Gangdong), and Xianju (1500 MW, Zhejiang) [13–15]. Moreover, more than 20 pumped-storage power stations are under construction in other 10 provinces in China.

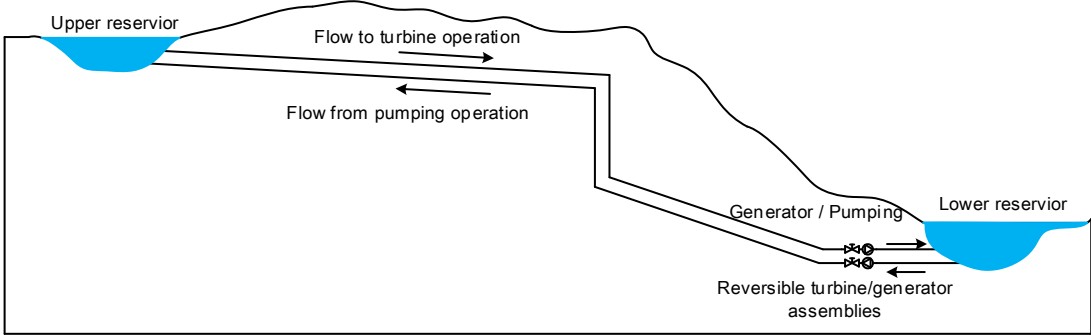

**Figure 1.** A pumped hydropower station layout.

**Table 1.** Major pumped-storage power stations completed in China.

| No. | Project | Location | Installed Capacity (MW) | Completed Year | Water Head (m) |
|---|---|---|---|---|---|
| 1 | Bailianhe | Hubei | 1200 | 2010 | 195 |
| 2 | Heimifeng | Hunan | 1200 | 2010 | 295 |
| 3 | Pushihe | Liaoning | 1200 | 2011 | 308 |
| 4 | Baoquan | Henan | 1200 | 2011 | 510 |
| 5 | Xiangshuijian | Anhui | 1000 | 2012 | 190 |
| 6 | Xianyou | Fujian | 1200 | 2013 | 640 |
| 7 | Huhehaote | Neimenggu | 1200 | 2015 | 513 |
| 8 | Qingyuan | Guangdong | 1280 | 2016 | 503 |
| 9 | Hongpin | Jiangxi | 1200 | 2016 | 540 |
| 10 | Xianju | Zhejiang | 1500 | 2016 | 440 |

　　Seepage control is one of the key technical issues in deep buried hydraulic tunnel and large-scale powerhouse of a pumped-storage power station. In those abovementioned pumped-storage power stations, concrete reinforced hydraulic tunnels were usually adopted, which are subjected to a maximum hydrostatic pressure of more than 2.0 MPa on the inner surface. The water pressure will inevitably induce tensile cracks in the concrete linings, and Table 2 lists the amount of leakage out of concrete reinforced hydraulic tunnels in the pumped-storage power stations in China [16–19], U.S. [20] and Norway [21], which results in enormous power loss day by day [22]. Consequently, there is an urgent need in understanding the nonlinear groundwater flow behaviors through porous and fractured rock, which is commonly encountered in various engineering applications, such as

underground tunneling, geothermal energy extraction, hydrocarbon production, and hazardous wastes isolation [23–25]. For instance, non-Darcy flow behavior in the near-well region was simulated with a nonlinear finite element model by Zhang and Xing [23]. A numerical model of three-dimensional unstable and nonlinear seepage was established, and nonlinear seepage movement characteristics of fluid in underground coal gasification (UCG) were studied by Yang [24]. Samanta et al. conducted direct numerical simulations of the fully turbulent flow through a porous square duct to study the effect of the permeable wall on the secondary cross-stream flow [26]. Li et al. presented a systematic performance assessment of the seepage control system designed for the underground caverns by finite element numerical modeling, with particular concerns on quantitative determination of the excavation-induced variation in hydraulic conductivity of the surrounding rock masses [27]. However, under the high water pressure environment, nonlinear flow behaviors would occur in the surrounding rock of hydraulic tunnel, which is rarely reported by previous literatures.

**Table 2.** Leakage events occurred in the pumped-storage power stations.

| No. | Project | Country | Installed Capacity (MW) | Water Head (m) | Leakage (L/s) | Power Loss (kW) |
|---|---|---|---|---|---|---|
| 1 | Guangzhou | China | 1200 | 610 | 32 | 1687 |
| 2 | Huizhou | China | 1200 | 624 | 230 | 12,400 |
| 3 | Tianhuangping | China | 1800 | 680 | 11.2 | 658 |
| 4 | Baoquan | China | 1200 | 639.6 | 150 | 8289 |
| 5 | Bath County | U.S. | 2l00 | 385 | 486 | 16,166 |
| 6 | Tafjord | Norway | 440 | 780 | 4 | 270 |

Seepage flow modeling is a very important step before system design, simulation and optimization. Given the fact that the high water pressure in the concrete reinforced manifolds, surrounding rocks are confronted with seepage stability problem in the pumped storage power station. In this study, the equivalent continuum finite element method (FEM) is employed in this study. A nonlinear flow numerical model combining the Forchheimer nonlinear flow theory, the discrete variational inequality formulation of Signorini's type and an adaptive penalized Heaviside function is established, which is employed to the seepage analysis of the hydraulic tunnel and powerhouse surrounding rocks in the Yangjiang pumped-storage power station. Moreover, the permeability of the rock mass under high water pressure is determined by high pressure packer test and its approximate analytical model.

## 2. Establishment of Nonlinear Seepage Mathematical Model

### 2.1. Governing Equations

Given the fact that the high water pressure, nonlinear flow would occur in the concrete reinforced hydraulic tunnels, and this type of flow can be adequately described by the Forchheimer equation (Forchheimer 1901) [23]:

$$-\nabla h = \frac{\mu}{\rho_w g k} v + \frac{\beta}{g} v |v| \tag{1}$$

where $\nabla h$ is the pressure gradient (or pressure drop), $\mu$ the dynamic viscosity of the fluid, $\rho_w$ is the fluid density, $v$ is the average velocity, $k$ is the intrinsicpermeability, and $\beta$ is the nonlinear coefficient dependent on the properties of the fractured rock.

Because of inconsistency in definitions and thus in critical values, no widely accepted criterion for nonlinear flow in porous media is available. The Forchheimer number $F_0$ is defined as the ratio of nonlinear (the second term in Equation (1)) to linear (the first term in Equation (1)) pressure losses in the Forchheimer's law, and it represents the ratio of the pressure gradient required to overcome inertial

forces to that of viscous forces. The Forchheimer number $F_0$ is adopted as criterion for nonlinear flow in this study because of the clear physical meaning of variables involved.

$$F_0 = \frac{k\beta\rho_w|\boldsymbol{v}|}{\mu} \tag{2}$$

Assuming the fractured rock is non-deformable and the fluid has a constant effective compressibility, substituting Equation (2) into (1) yields.

$$-\nabla h = \frac{\mu}{\rho_w g k}\left(1 + \frac{k\beta\rho_w|\boldsymbol{v}|}{\mu}\right)\boldsymbol{v} = \frac{\mu}{\rho_w g k}(1 + F_0)\boldsymbol{v} \tag{3}$$

As demonstrated in Figure 2, seepage flow through rock domain $\Omega = \Omega_w \cup \Omega_d$ is then governed by the following equation of continuity.

$$\nabla \cdot \boldsymbol{v} = 0 \quad (in\ \Omega) \tag{4}$$

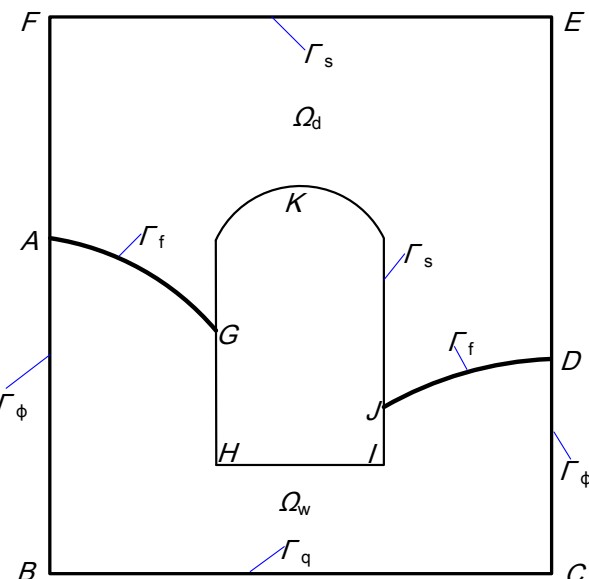

**Figure 2.** Sketch of seepage flow through an underground cavern.

Starting from Equations (3) and (4) and following the discrete variational inequality formulation of Signorini's type proposed by Chen et al. [28], one obtains the governing equation for nonlinear flow in fractured rock equation:

$$\nabla \cdot \left(K\frac{1}{1 + F_0}\nabla h\right) = 0 \tag{5}$$

in which $K = \mu\frac{k}{\rho_w g}$ is the permeability of the fractured rock.

Equation (5) is subjected to the following boundary conditions:

- The water head boundary condition

$$\phi = \overline{\phi} \tag{6}$$

where $\overline{\phi}$ is the prescribed water head on $\Gamma_\varphi$, $\phi = z + p/\gamma_w$ is the total water head, $z$ the vertical coordinate, $p$ the pore water pressure, $\gamma_w$ the unit weight of water.

- The flux boundary condition

$$q_n \equiv -\boldsymbol{n}^{\mathrm{T}}\boldsymbol{v} = \overline{q} \tag{7}$$

where $\bar{q}$ is the prescribed flux on $\Gamma_q$, and $\boldsymbol{n}$ the outward unit normal vector to the boundary. For an impermeable boundary, $\bar{q} = 0$.

- The boundary condition of Signorini's type on the seepage surface

$$\begin{cases} \phi \leq z, \; q_n(\phi) \leq 0 \\ (\phi - z) \, q_n(\phi) = 0 \end{cases} \tag{8}$$

where $\Gamma_s$ is the potential seepage boundary. Obviously, on section GHIJ, $\varphi = z$ and $q_n \leq 0$; while on sections AFED and GKJ, $\varphi < z$ and $q_n = 0$. $\varphi = z$ and $q_n = 0$ are satisfied at seepage points G and J.

- The boundary condition on the phreatic surface

$$q_n|_{\Omega_w} = q_n|_{\Omega_d} = 0 \tag{9}$$

where $\Gamma_f \equiv \{(x, y, z \,|\, \varphi = z)\}$ is the phreatic surface, an interface between $\Omega_w$ and $\Omega_d$.

### 2.2. Solution Procedure and Finite Element Numerical Methods

The mathematical statement for a discrete version of the iterative formulation is given as follows: find a vector $\boldsymbol{\Phi}_{VI}^h = \left\{ \boldsymbol{\phi} \,|\, \boldsymbol{\phi} \in R^j; \phi_i = \overline{\phi}_i, \text{ for } i \in \Gamma_\phi; \quad \phi_i \leq z_i, \text{ for } i \in \Gamma_s \right\}$, such that for $\forall \boldsymbol{\psi} \in \boldsymbol{\Phi}_{VI}^h$, the following inequality holds:

$$(\boldsymbol{\psi} - \phi^{l+1})^\mathrm{T} \boldsymbol{N} \phi^{l+1} \geq (\boldsymbol{\psi} - \phi^{l+1})^\mathrm{T} q^l \tag{10}$$

with

$$\boldsymbol{N} = \sum_e \iiint \Omega_e \boldsymbol{B}^\mathrm{T} \frac{1}{1 + F_0} K \boldsymbol{B} \mathrm{d}\Omega \tag{11}$$

$$q^l = \sum_e \iiint \Omega_e \boldsymbol{B}^\mathrm{T} v_0^l \mathrm{d}\Omega = \boldsymbol{N}_\varepsilon \boldsymbol{\phi}^l \tag{12}$$

$$\boldsymbol{N}_\varepsilon = \sum_e \iiint \Omega_e \boldsymbol{H}_\lambda (\phi^m - z) \boldsymbol{B}^\mathrm{T} \frac{1}{1 + F_0} K \boldsymbol{B} \mathrm{d}\Omega \tag{13}$$

where $l$ is the iterative step, $j$ the total number of nodal points in the finite element mesh, $\boldsymbol{B}$ the geometrical matrix of the finite element model, and $H_\lambda$ the adaptive penalized Heaviside function introduced to evade numerical instability and mesh dependency, given by Chen et al. [28].

$$H_\lambda(\phi - z) = \begin{cases} 1 & \text{if } \phi \leq z - \zeta\lambda_1 \\ \frac{z + \zeta\lambda_2 - \phi}{\zeta(\lambda_1 + \lambda_2)} & \text{if } z - \zeta\lambda_1 < \phi < z + \zeta\lambda_2 \\ 0 & \text{if } \phi \geq z + \zeta\lambda_2 \end{cases} \tag{14}$$

where $\lambda_1$ and $\lambda_2$ are two parameters associated with each element. The parameter $\lambda_1$ is defined as the vertical distance between the lowest integration point and the lowest node, and $\lambda_2$ the vertical distance between the highest integration point and the highest node, in the FEM model concerned. The symbol $\zeta$ is introduced to scale values of parameters $\lambda_1$ and $\lambda_2$, so that stronger convergence criteria could be ensured for highly nonlinear problems with coarse meshes. Moreover, the introduction of $\zeta$ in the suggested range does not significantly influence the behavior of the penalized Heaviside function, but results in very good numerical convergence [28].

The discretized PDE formulation given in Equation (10) was implemented in a FEM code, THYME3D, initially developed for coupled deformation/multiphase flow/thermal transport analysis in geological porous media [29].

In the iterative process, the following convergence criteria is used:

$$\|\phi^{m+1} - \phi^m\|_1 \leq \varepsilon_1 \|\phi^m\|_1 \text{ and } \|\phi^{m+1} - \phi^m\|_2 \leq \varepsilon_2 \|\phi^m\|_2 \tag{15}$$

in which $\phi^l$ and $\phi^{l+1}$ denote the water head at the $l$ and $l+1$ step, respectively. The symbols $\varepsilon_1$ and $\varepsilon_2$ denote the user-specified error tolerances and take the values of $\varepsilon_1 = 10^{-3}$ and $\varepsilon_2 = 10^{-4}$ in this study.

## 3. Site Characterization of the Yangjiang Pumped-Storage Power Station

### 3.1. General Description and Geological Condition

Yangjiang pumped-storage power station is under construction in Yangjiang County, Guangdong Province, China, which includes an upper reservoir, a lower reservoir and hydropower system with installed capacity and annual output were 2400 MW and 3432GWh, respectively. The upper reservoir is located in Efengling provincial nature reserve area, with effective catchment area of 7.54 km² and normal pool level 773.7 m. The lower reservoir is located in Litian River, with effective catchment area of 15.94 km² and normal pool level 103.7 m. As illustrated in Figure 3, the diversion tunnel, 7.4 m in diameter, is composed of an upper horizontal section, a middle horizontal section, a lower horizontal section and two inclined sections that connect the horizontal ones. At the lower horizontal section, the hydraulic tunnel is branched into three smaller tunnels of 3.0 m in diameter, which carry water to operate three reversible turbine/generator assemblies installed in the hydropower. The tunnels are concrete reinforced, and at the lower horizontal section, they are subjected to a maximum hydrostatic pressure of 7.99 MPa and an extra surge pressure up to 3.0 MPa on the inner surface in the condition of general operating.

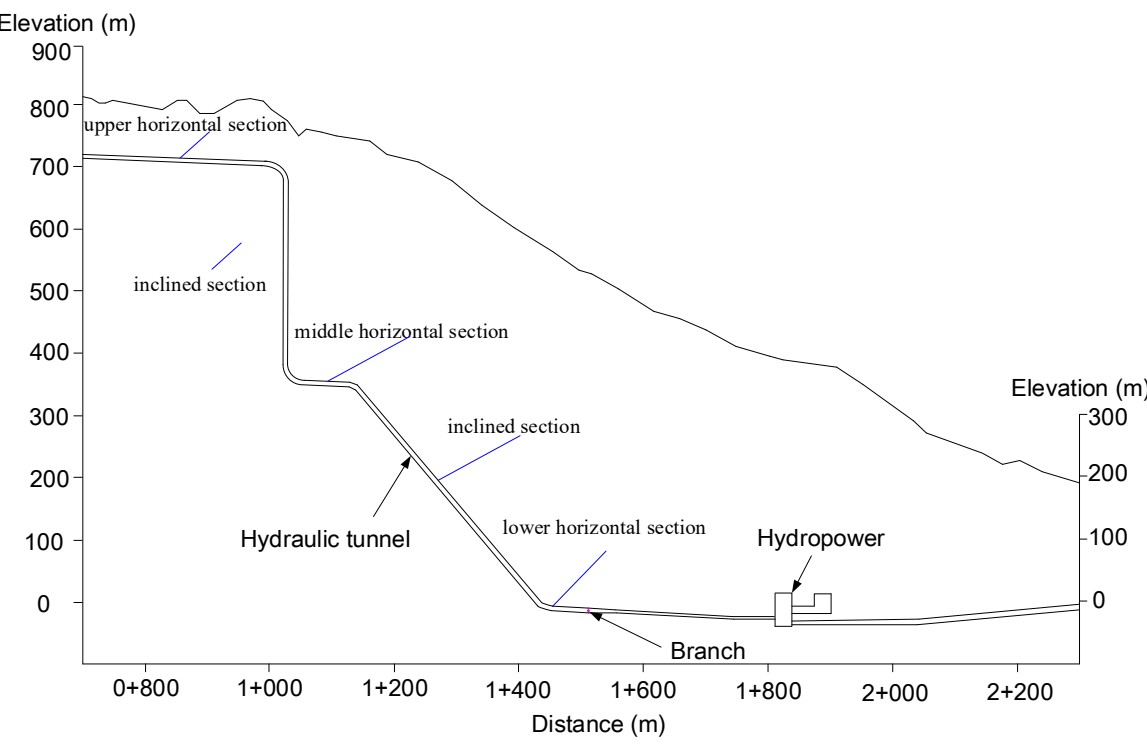

**Figure 3.** Layout of the diversion tunnel in the Yangjiang pumped-storage power station.

The bedrocks in the study area mainly contain sedimentary rocks of the Cretaceous Lumuwan group ($K_{1Lm}$) and Indosinian granites ($\eta\gamma_5^{1-3}$), with $K_{1Lm}$ unconformably overlying $\eta\gamma_5^{1-3}$. The Cretaceous Lumuwan group ($K_{1Lm}$) is abundantly distributed in the lower reservoir area, and it consists of sandstone and conglomerate. The Indosinian granites ($\eta\gamma_5^{1-3}$) are mostly distributed in the upper reservoir area, and consist of chloritization granite, biotite granite and medium-grained granite. As demonstrated in Figure 4, the main structures at the site consist of five sub-vertically oriented faults, f721, f717, f715, f708 and f745, with f721 striking towards N70°~75°W, N75°W, N85°E, N10°W, and

N5°~15°E, respectively. Fault f721, f717, f715, f708 and f745 is about 0.1–1.2 m in width and extends for about 700 m, and details of these faults are listed in Table 3.

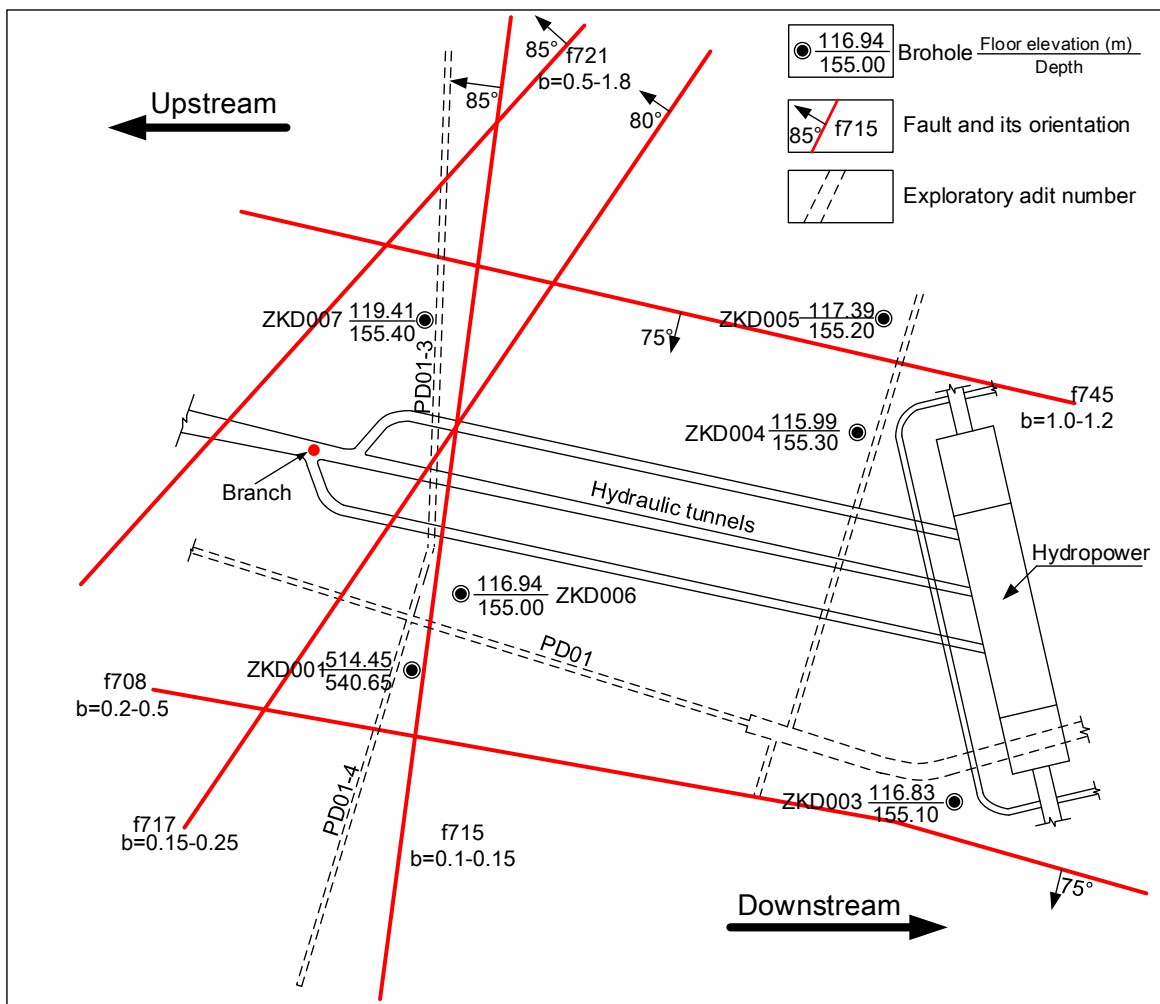

**Figure 4.** Layout of the geological map in the Yangjiang pumped-storage power station.

**Table 3.** Key faults information selected in the study area.

| No. | Occurrence | Width (m) | Fault Characteristic |
|---|---|---|---|
| f721 | N70~75W/SW∠85° | 0.5~2.0 | Silicified cataclastic rock powder, crushed rock, mylonite, partial filling 0.5~1.0 cm Quartz vein, weak cementation generally-well medium silicide, linear leaks, $Q$ = 2~3 L/min. |
| f717 | N75W/SW∠80° | 0.15~0.25 | Silicified cataclasite, cementation general well, impact zone 0.5~1.0 m, Linear strands of leakage, $Q$ = 4~5 L/min. |
| f715 | N85E/SE∠85° | 0.1~0.15 | Silicified cataclasite, fractured fault, mild alteration zone 0.2~0.3 m, Linear leaking, $Q$ = 1 L/min. |
| f708 | N10W/NE∠70~75° | 0.2~0.6 | Cataclasite and breccia, mylonite fault gouge, cementation, quartz vein invasion, linear, form water, $Q$ = 15~20 L/min. |
| f745 | N5~15E/SE∠70~80° | 0.2~1.2 | Cataclastic rock, silicified rock, cementation, permeability drop~linear strip break water, $Q$ = 0.4~0.5 L/min. |

### 3.2. Proposed Seepage-Control Measures

Figure 5 illustrates seepage-control measures for the hydropower in the Yangjiang pumped storage power station, which is composed of high pressure consolidation grouting, impervious curtain, galleries and drainage holes. To form a complete impervious system in the branch of hydraulic tunnels, the high pressure consolidation grouting is arranged around the hydraulic tunnels, with 6.0 m in depth, 230 m in length and 1.3 times as the static head high in grouting pressure. The "V" shaped curtain grouting galleries (A1) is arranged right above the consolidation grouting, and the curtain extends 2.0 m and 3.0 m respectively from the upstream and downstream hydraulic tunnels. The proposed space between the consolidation grouting and curtain grouting holes is 2 m, with 76mm in the diameter. At the same time, the drainage galleries are deployed in the end of the hydraulic tunnels (A2) and in the surrounding rock of the hydropower (A3~A6) to form a drainage system. Drainage holes are vertically deployed in the surrounding rock immediately behind the grouting curtain, and the drainage galleries close to the hydropower are connected by the vertical drainage holes. The drainage holes are deployed with 5 m in spacing and 120.0 m in depth, and the size of the cross-sections of the holes is assumed to be 12.0 cm in diameter.

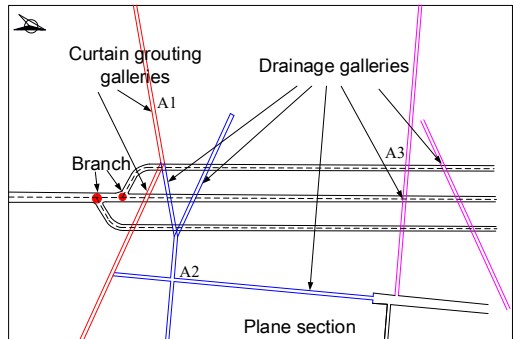 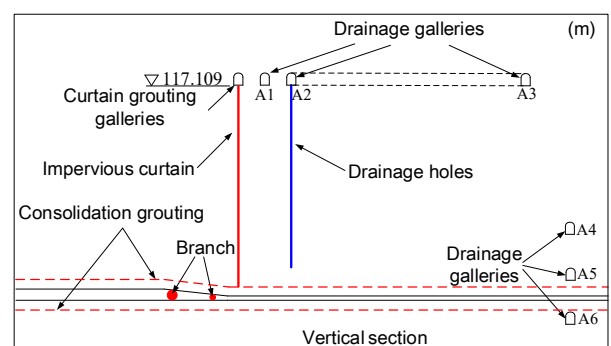

**Figure 5.** Illustration of seepage-control measures for the hydropower in the Yangjiang pumped storage power station.

## 4. Numerical Modeling of the Nonlinear Seepage in the Yangjiang Pumped-Storage Power Station

### 4.1. The Finite Element Model

To simulate the nonlinear seepage filed and assess the performance of the seepage-control measures in the surrounding rock of the hydraulic tunnels and hydropower, a 3-D finite element (FE) mesh, with 455,110 brick elements and 316,464 nodes in total, was generated, as shown in Figure 6. The size of the mesh is 460 m along the pumping or generating hydraulic tunnels water flow direction, and 360 m in another horizontal direction. According to Chen et al. [28], for the proposed parabolic PDE and the discrete version, the accuracy of numerical simulation is dependent on the mesh density. Therefore, to improve the calculation accuracy of the seepage field around the hydraulic tunnels and water outflow, the grid around the hydraulic tunnels was more intensive. Simultaneously, the topographic and geological features at the study area, and the seepage-control measures containing high pressure consolidation grouting, impervious curtain, galleries and drainage holes.

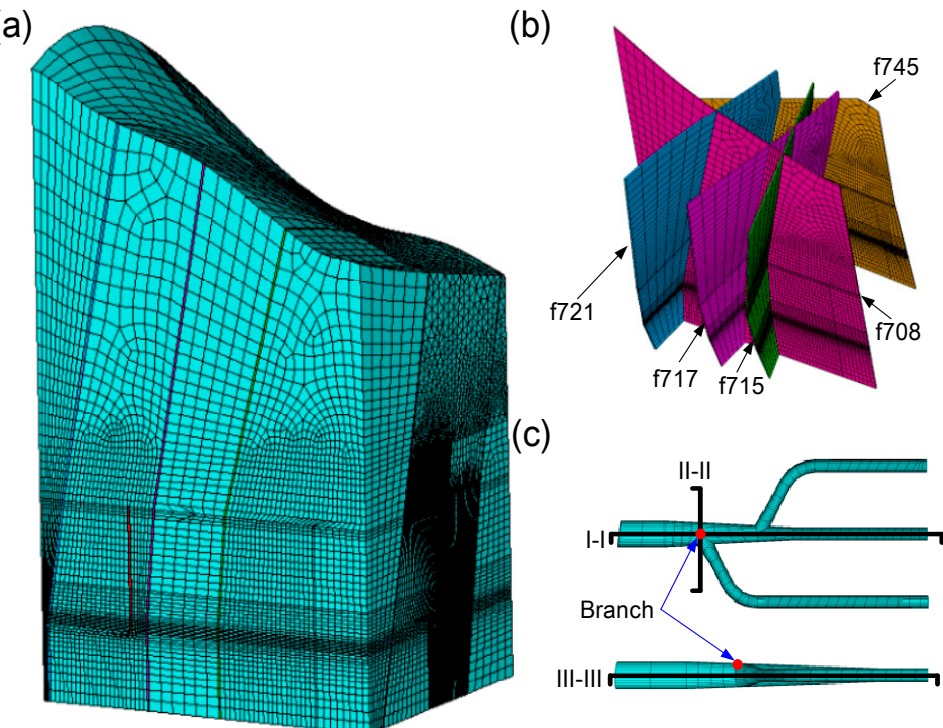

**Figure 6.** 3-Dfinite element (FE) of the study area: (**a**) finite element, (**b**) key faults, and (**c**) hydraulic tunnels.

### 4.2. The Calculation Parameters

To obtain the permeability of the surrounding rock under high water pressure, high pressure packer test (HPPT) was conducted of the boreholes in the study area, as plotted in the Figure 4. Thereafter, on the basis of Izbash's nonlinear empirical equation, an approximate analytical model was developed especially for interpreting data from these HPPTs.

Assuming the volumetric flow rate ($q$) injected into the infinitesimal section to be linearly proportional to the volumetric total flow rate ($Q$) injected into the test interval ($L$) yields

$$q = \frac{Q}{L} \tag{16}$$

Further, assuming radial velocity $v_\rho$ of the test interval yields

$$v_\rho = \frac{q}{2\pi\rho} = \frac{Q}{2\pi L\rho} \tag{17}$$

where $\rho$ is the radial distance from the borehole axis.

On the plane vertically intersecting the center of the test interval, the Izbash's law is similarly written as

$$-\frac{\partial P_r}{\partial r} = \lambda(v_r)^m = \lambda(v_\rho)^m \tag{18}$$

where $P_r$ is the water pressure at the distance $r$ from the borehole, $v_r$ the tangential velocity, and $\lambda$ and $m$ is empirical coefficient in the Izbash's law. For laminar flow ($m = 1$), the Izbash's law reduces to the Darcy's law, and for $m = 2$, Equation (18) represents a fully turbulent flow. In transitional flow condition, the value of $m$ ranges from 1 to 2.

Beyond the radius of influence $R_0$, the pressure is assumed to vanish

$$\lim_{r \geq R_0} P_r = 0 \tag{19}$$

Substituting Equation (17) into (18) and integrating Equation (18) yields

$$P_r = \int_r^{R_0} -\frac{\partial P_r}{\partial r} dr = \int_r^{R_0} \lambda (v_r)^m dr = \int_r^{R_0} \lambda \left(\frac{Q}{2\pi L \rho}\right)^m d\rho \tag{20}$$

The analytical solution for the 2D radial flow is then derived as

$$
\begin{cases}
m = 1 & P_r = \lambda \cdot \frac{Q}{2\pi L \rho} \cdot \ln\left(\frac{R_0}{r}\right) \\
m \neq 1 & P_r = \lambda \cdot \left(\frac{Q}{2\pi L \rho}\right)^m \cdot \left\{\frac{r^{1-m} - R_0^{1-m}}{m-1}\right\}
\end{cases}
\tag{21}
$$

Rearranging Equation (21) yields the expression for permeability *k*:

$$
\begin{cases}
m = 1 & k = \frac{Q}{2\pi L P_r} \cdot \ln\left(\frac{R_0}{r}\right) \\
m \neq 1 & k = \frac{Q}{2\pi L P_r^{1/m}} \cdot \left\{\frac{r^{1-m} - R_0^{1-m}}{m-1}\right\}^{1/m}
\end{cases}
\tag{22}
$$

Based on the above mentioned in situ tests and approximate analytical model, permeability of the surrounding rock is listed in Table 4. Moreover, permeability of high pressure consolidation grouting and impervious curtain is $3.68 \times 10^{-6}$ and $5.0 \times 10^{-6}$ cm/s, respectively.

**Table 4.** Permeability of the strata and faults at the study area (unit: cm/s).

| Strata | Normal Permeability | Tangential Permeability |
|---|---|---|
| Intact surrounding rock | $1.26 \times 10^{-6}$ | $1.26 \times 10^{-6}$ |
| Relatively intact surrounding rock | $3.68 \times 10^{-6}$ | $3.68 \times 10^{-6}$ |
| Fractured surrounding rock | $3.84 \times 10^{-5}$ | $3.84 \times 10^{-5}$ |
| f721 and f745 | $3.68 \times 10^{-6}$ | $2.0 \times 10^{-4}$ |
| f708, f715 and f717 | $3.68 \times 10^{-6}$ | $5.0 \times 10^{-4}$ |

### 4.3. The Boundary Conditions

The back analysis showed that when the water head on the upper reservoir stakes a normal water level of 773.7 m, the objective function was minimized over all the boreholes with available groundwater level measurements. Under this condition, the upstream of the FEM model water level is 770 m, and the corresponding water level in the downstream is 208.5 m. The lateral boundary at the mountain side and the base boundary of the model was assumed to be impermeable. The ground and dam surfaces above the upstream and downstream water levels, the surfaces of the hydraulic tunnels, the drainage galleries and the drainage holes, are all taken as the potential seepage boundaries satisfying the Signorini's complementary condition.

### 4.4. Performance Assessment of the Seepage-Control Measures

Figure 7 shows the distribution of phreatic surface and pressure head at cross-section I-I, with its location shown in Figure 6c. One observes from Figure 7 that the pressure head around the surrounding rocks of the hydraulic tunnels is rather high due to the low-permeability high pressure consolidation grouting around the tunnels. Moreover, the closer to the branch of the hydraulic tunnels, more intensive distribution of the pressure head. Both abovementioned results showed that the high pressure consolidation grouting plays a very good anti-seepage effect. Figure 8 shows the distribution of phreatic surface and pressure head at cross-section II-II, with its location shown in Figure 6c. It demonstrates that the impervious curtain at the upstream of the branch, and the drainage facilities and holes has a dramatic impact on the seepage flow and can sharply depress the free surface, resulting in a remarkable fall of external water pressure above the hydraulic tunnels. These phenomenaare also observed in Figure 9, where water pressure decreases to about 200 m after

the seepage-control measures around the hydraulic tunnels. The simulation results show that the suggested seepage-control measures is effective in lowering the groundwater level and reducing the pore water pressure in the surrounding rocks around the hydraulic tunnels, and the possible concentrated flow through Faults f721, f745, f708, f715 and f717 is effectively avoided.

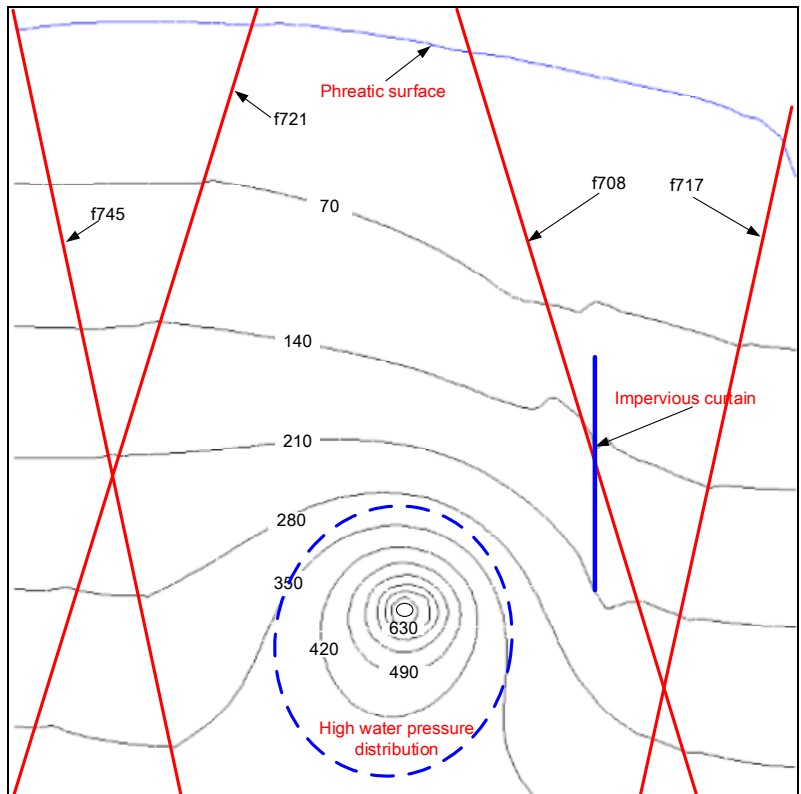

**Figure 7.** Phreatic surfaces and pressure distribution in the surrounding rocks at cross-section I-I.

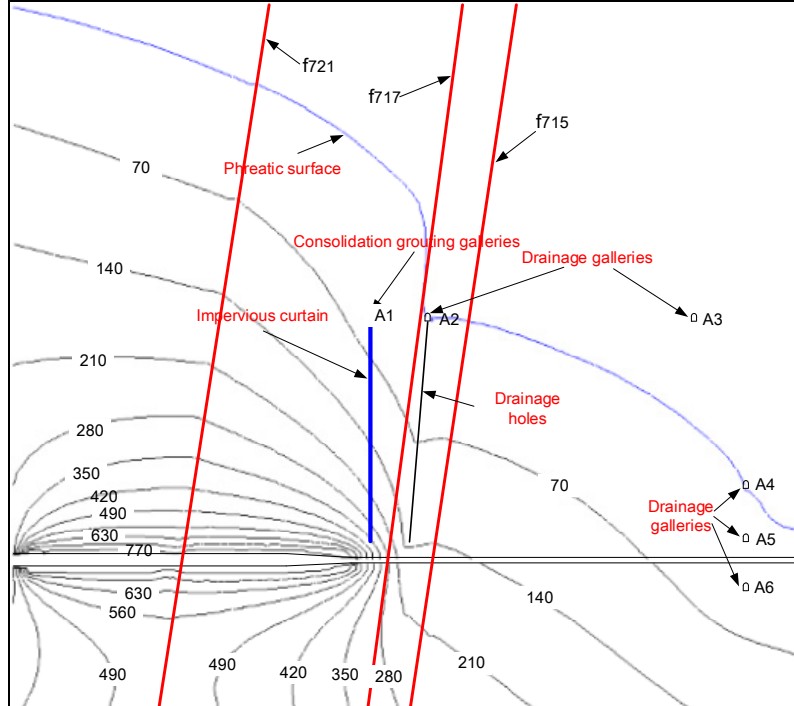

**Figure 8.** Phreatic surfaces and pressure distribution in the surrounding rocks at cross-section II-II.

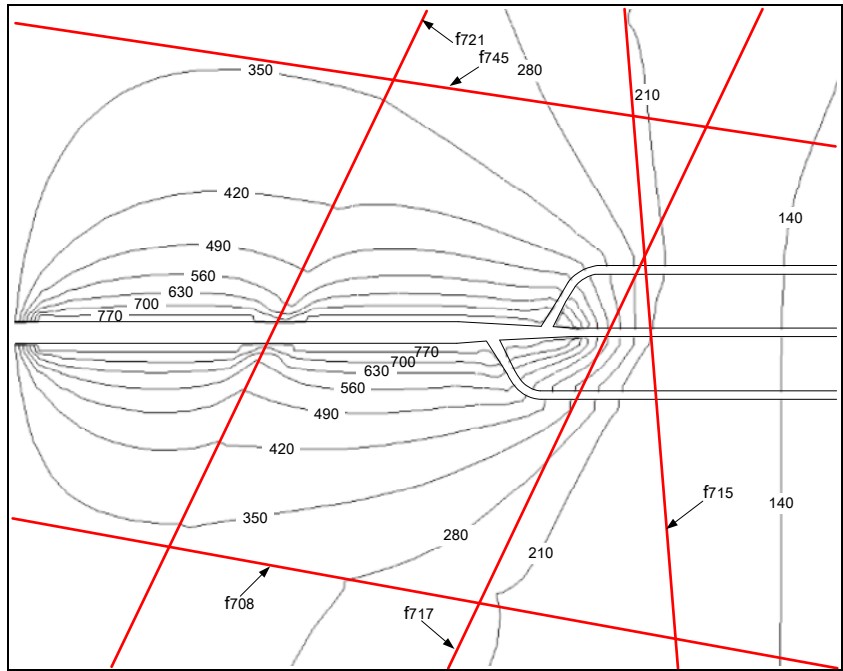

**Figure 9.** Phreatic surfaces and pressure distribution in the surrounding rocks at cross-section III-III.

### 4.5. Nonlinear Seepage Characteristic in the Surrounding Rocks of the High Water Pressure Hydraulic Tunnels

To illustrate the nonlinear flow characteristic in the surrounding rocks of the hydraulic tunnels, Figures 10 and 11 plot the distribution of the Forchheimer number $F_0$ at cross-section of the A2 and A3 drainage galleries, respectively. One observes from Figure 10 that the nonlinear seepage mainly occurs in the areas nearby faults and seepage-control measures. The maximum values were 0.7 and 1 at the bottom of drainage holes and upstream of A3 drainage galleries, where the nonlinear flow is the most strongly. That is because the hydraulic gradient in the surrounding rocks near the seepage-control measures are so large, which would lead nonlinear flow easily. In addition, special attention must be focused on the fault f708 and f717, where the high water pressure is firstly occurs at the f708 and rapidly extend to f717. Then, a potential seepage channel will growth, which may result the leakage in the surrounding rocks of the hydraulic tunnels. In order to demonstrate the nonlinear effect on water inflow and outflow, the flow rate at the branch of the hydraulic tunnels, A1–A6 drainage galleries and drainage holes are calculated by the linear and nonlinear seepage theories.

As listed in Table 5, the leakage is calculated by nonlinear flow theory are significantly less than the linear flow, such as leakage are 32.9 and 40.5 L/s at the branch of the hydraulic tunnels, and corresponding power loss are 2189 and 2694 kW·h, respectively. Moreover, the difference is getting bigger as the nonlinear flow stronger, and this is because the calculated flow velocity by nonlinear flow theory is smaller than the linear theory.

**Table 5.** Comparison of the flow rate for linear and nonlinear flow theories (L/min).

| Location | Water Outflow | Water Inflow | | |
|---|---|---|---|---|
| | Branch of the Hydraulic Tunnels | A1–A3 Drainage Galleries | Drainage Holes | A4–A6 Drainage Galleries |
| Linear flow | 40.5 | 17.2 | 12.9 | 1.2 |
| Nonlinear flow | 32.9 | 14.5 | 10.5 | 1.2 |

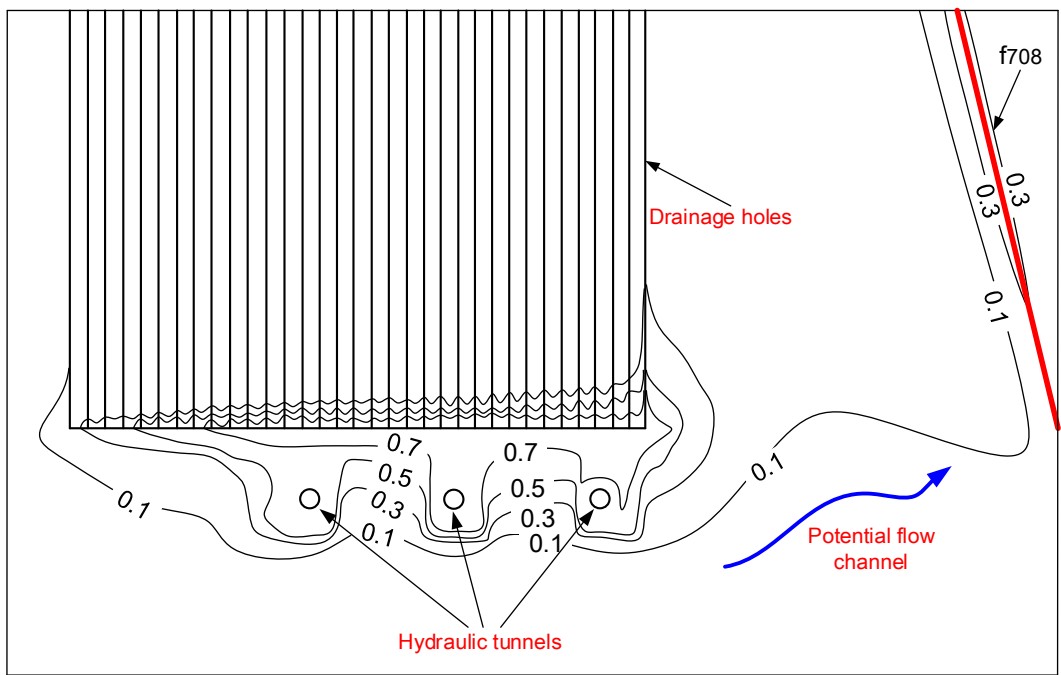

**Figure 10.** The $F_0$ contours at cross-section of the A2 drainage galleries.

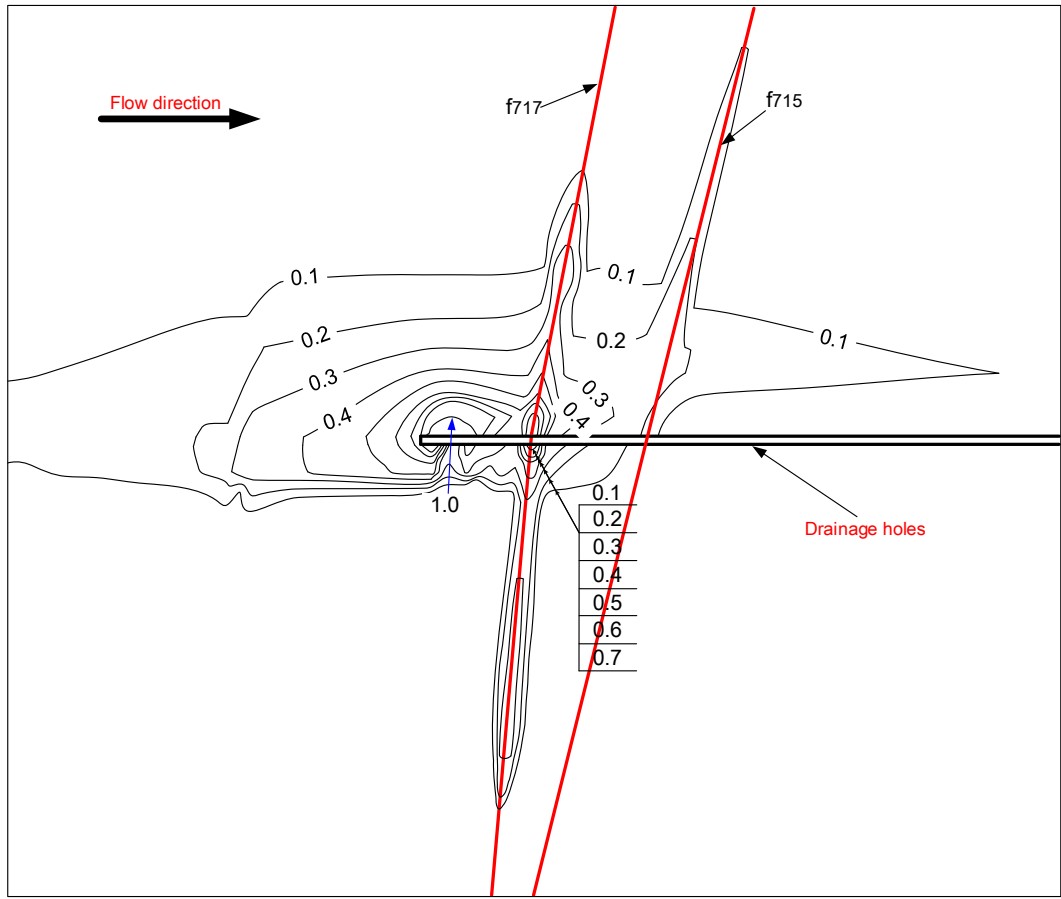

**Figure 11.** The $F_0$ contours at cross-section of the A3 drainage galleries.

## 5. Conclusions

A nonlinear seepage numerical model is built in this study for numerical simulation the nonlinear seepage in a pumped-storage power station. This model contains the Forchheimer nonlinear flow theory, the discrete variational inequality formulation of Signorini's type and an adaptive penalized Heaviside function, which is employed to demonstrate the performance assessment of the seepage-control measures and nonlinear flow characteristic in the surrounding rocks of the hydraulic tunnels. The major conclusions of this case study are summarized as follows:

(1) The simulation results show that the suggested seepage-control measures is effective in lowering the groundwater level and reducing the pore water pressure in the surrounding rocks around the hydraulic tunnels, and the possible concentrated flow through Faults f721, f745, f708, f715 and f717 is effectively avoided.

(2) Flow in the surrounding rocks is particularly prone to become nonlinear as a result of the high flow velocities and hydraulic gradients in the nearby of the seepage-control measures and the high permeability fault. The flow rate derived from nonlinear flow theory is smaller than the Darcy flow theory, what is more remarkable as the increase of nonlinear flow. Therefore, it is necessary to adopt nonlinear flow theory in the seepage analysis in the condition of high water pressure.

(3) With the development of the pumped storage power stations in China, the impervious design and safety control of high head hydropower projects are the key technical problems that need to be solved urgently. Preliminary work on the nonlinear seepage analysis of surrounding rock has carried out in this study, more efforts are needed in the theory, analysis method and control measures of nonlinear seepage in geotechnical engineering.

**Author Contributions:** S.H. confirmed the series of simulation parameters and arranged and organized the entire simulation process. X.Z. checked and discussed the simulation results. Y.L. made many useful comments and simulation suggestions. G.Z. supervised this research and reviewed the manuscript.

**Funding:** This study supported by the financial supports from the National Natural Science Foundation of China (No. 51179136) and the Hubei Key Laboratory of Roadway Bridge and Structure Engineering (Wuhan University of Technology) (No. DQJJ201705).

**Conflicts of Interest:** The authors declare no conflict of interest.

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
