# Peer review of "Numerical Simulation Three-Dimensional Nonlinear Seepage in a Pumped-Storage Power Station: Case Study"

_energies, doi:10.3390/en12010180_

Round 1

Reviewer 1 Report

The manuscript is very interesting but I think the authors should show this interest.

They have to describe in the introduction what is the energy cost due to leakeage in both modes (turbine and pump) iin the review literature

They have to enumarete the used software as well as they could do a calibration (they or other researches which did similar simulations)

They have to calculate the energy cost in their case stuy

They have to propose the application of this model in other systems. or it is only useable in this station.

Author Response

Dear Reviewer,

Thanks much for the comments. Following all of the comments, we have made a great effort to address the comments and to improve the English writing. A response list to the comments is attached below. All the changes were marked in red in the revised manuscript. We hope the revised version would meet the high quality of the journal.

Yours sincerely,

Xinlong Zhou

Reviewer 2 Report

Good work. Observations:

General observation:

Be more extensive when describing the hydropower development. Introduce better table 5 with final results of the calculations, based on what you draw the conclusions.

Punctual observations:

Reconsider the utility of figure 1 in a scientific article.

Use the same font for the average velocity, v, in all the equations.

Review equations 4, 14.

Rename figure 2 and explain the notations.

Describe the setup of the PSP a little bit more in paragraph 3.1 (the three hydraulic tunnels…), in order to link better with paragraph 3.2.

Rename figure 3, add title to horizontal axis and explain the notations.

In eq. 1 Greek letter ro is the fluid density (Lines 108-109), in eq. 17, ro is the radial distance from the borehole axis (Line 232). Solve this superposition of notations.

Explain m from eq. 18.

Lines 314-315: This statement appears often: The flow rate calculated by nonlinear flow theory is less than the Darcy flow 314 theory, and the difference is getting bigger as the nonlinear flow stronger. Please develop a little bit more this conclusion.

Author Response

(The authors gave the same response as above.)

Reviewer 3 Report

This work discusses a very interesting numerical analysis with application to the improvement of hydropower plant operation. After addressing the following points, the article would be suitable for publication:

Make sure that everything in equations (10)-(14) is defined. For instance, what is K_epsilon?

Also, try to have a unique set of symbols throughout the paper. For example, in equations (1), k is the intrinsic permeability and in equation (10) the same symbol is the iterative step.

Regarding the station described in section 3.1, how representative is it of general operating conditions?

The introduction reflects that this is and interesting analysis and it could be connected to other areas where the management of a variable renewable resource is critical for optimum energy usage. For instance, the example of concentrated solar power (CSP), as in the work by Cachafeiro et al. (Energy Procedia 69, 299-309, 2015).

The scope of this work could be connected to the existing high-fidelity simulations of complex flows (such as porous ducts), which can bring some additional insight on the underlying physics. See for instance the work by Samanta et al. (J. Fluid Mech. 784, 681-693, 2015).

Figures 2 and 3 are in some sense redundant. If both are necessary, at least change the caption.

Section 4.1: what was the mesh design strategy? Did the authors conduct any grid independence tests?

There are some typos that could be fixed throughout the manuscript.

Author Response

(The authors gave the same response as above.)

Round 2

Reviewer 1 Report

The authors did the proposed changes and the document improved consideribly.